# Analysis of the Airflow Generated by Human Activity Using a Mobile Slipstream Measuring Device

**Minkyeong Kim** [1], **Yongil Lee** [2] **and Duckshin Park** [3],*

1    Railroad Test & Certification Division, Korea Railroad Research Institute, Uiwang 16105, Korea; mkkim15@krri.re.kr
2    Environmental Monitoring Group, Han-River Basin Environmental Office, Hanam 12902, Korea; freego83@korea.kr
3    Transportation Environmental Research Department, Korea Railroad Research Institute, Uiwang 16105, Korea
*    Correspondence: dspark@krri.re.kr

**Abstract:** Human activities, including walking, generate an airflow, commonly known as the slipstream, which can disperse contaminants indoors and transmit infection to other individuals. It is important to understand the characteristics of airflow to prevent the dissemination of contaminants such as viruses. A cylinder of diameter 500 mm, which is the average shoulder width of an adult male, was installed in a motorcar and moved at a velocity of 1.2 m/s, which is the walking speed of an adult male. The velocity profile of the slipstream generated during this movement was measured by locating the sensor support at 0.15–2.0 m behind the cylinder. The wind velocity was set to 1.2 m/s to conduct the numerical analysis. The measurement data revealed the velocity profile of the space behind the cylinder, and a comparison of the numerical analysis and the measurement results indicate very similar $u$ (measured velocity)/$U$ (moving velocity) results, with a maximum difference of 0.066, confirming that the measured values were correctly estimated from the results of the numerical analysis.

**Keywords:** infectious viruses; indoor contaminants; mobile slipstream measuring instrument; cylinder; airflow

## 1. Introduction

The severe acute respiratory syndrome epidemic that began in Hong Kong in 2003 reportedly spread widely through the air [1]. Middle East respiratory syndrome (MERS) was first reported in Saudi Arabia in 2012 and then spread worldwide, including Korea. Severe acute respiratory syndrome coronavirus 2 (COVID-19) was first detected in December 2019, in Wuhan City, Hubei Province, China, and then very quickly spread throughout other countries including Korea. Although the transmission pathway of MERS and other pathogens has not been identified with 100% certainty, it is presumed that it is transmitted via close contact among human beings [2]. In this case, the term "close contact" is defined as being within 2 m of another person, in a room with others, or in the care area of an infected person; it also includes direct contact with infectious secretions while not wearing appropriate personal protective equipment [2]. Airflow generated by human movements can accelerate the spread of various airborne materials, such as COVID-19 virus particles during the current pandemic [3–7].

Numerical analyses are used by many researchers to understand aerodynamic properties because the procedures have no spatial constraints. Regarding the dissemination of substances, airflow by heating, ventilation, and air conditioning systems have been reviewed [3–7]. The main factors in the diffusion of pollutants are the airflow generated by human breathing and body heat [8,9]. Liu et al. [8] numerically analyzed natural convection, and Martinho et al. [9] modeled an actual mannequin using 3D scans to analyze the airflow generated by body heat when the mannequin was placed in a sitting position

indoors. In both studies, the results were compared with actual measurements. One study conducted a numerical analysis of airflow around the human body, taking into account the layer of air created by clothes [10]. The trajectory of particles exhaled by breathing was numerically analyzed according to the wind speed of the surrounding airflow [11]. Pollutant movement according to the airflow generated while passing through a door between a room and hallway was analyzed [12–15]. Simulations and numerical analyses have been performed using the speed of a moving person and the opening and closing of a door [16].

To evaluate the diffusion of pollutants, a study using a chamber was conducted. Zhang et al. [17] built a cabin with dimensions of 4.9 × 4.23 × 2.1 m in a chamber to evaluate the dissemination of contaminants in aircraft cabins. Experimental measurements and numerical simulations of airflow and contaminant transport were conducted in a "half occupied, twin-aisle cabin mockup". Poussou and Mazumdar [18] simulated an aircraft cabin using a small underwater tank and measured the variables using particle image velocimetry and planar laser-induced fluorescence. Contaminant propagation according to human movement in the aircraft cabin was analyzed [19]. Han et al. [20] built a test bed that could house actual-size mannequins and installed a wind velocity sensor to measure the velocity profile under different conditions. The airflow was measured using a warm mannequin and a mannequin with mobile parts moving at a velocity of 0.5–1.5 m/s. Milanowicz and Kedzior [21] modeled a human body falling from a height. Liu et al. [22] examined the airflow around a human body model in an enclosed space, examined changes in temperature and velocity, and simulated the transmission of infectious respiratory diseases.

Studies using mannequins have compared numerical analysis and measurement results, and confirmed particle diffusion [23,24]. In this study, a mobile slipstream measuring device was constructed to measure the velocity profile behind an object. The velocity profile behind a moving cylinder was measured directly, and a numerical analysis was performed to elucidate the airflow characteristics of the slipstream. In previous studies, no attempts were made to change the speed of the moving cylinder, or to detect changes in airflow by installing a flow rate sensor on the side of the cylinder.

Airflow is generated by human movements, which can affect the acceleration of the spread of various airborne materials, such as the COVID-19 virus, which is currently an important issue [3–7]. In this study, a moving cylinder was used to represent an adult male, and a constant walking pace was maintained. A regularly shaped cylinder of diameter 500 mm (average shoulder width of an adult male) was mounted on a motorcar and set to move at a velocity of 1.2 m/s (average walking velocity of humans). This cylinder approximates the movement of the human body. A flow rate sensor was installed on the back of the moving cylinder at different heights and distances to measure the change in airflow. In previous studies, no attempt was made to change the speed of a moving cylinder or to detect the changes in airflow by installing a flow rate sensor on the side of a cylinder. This would enable the changes in air flow around the cylinder to be determined during movement, and height and distance could be considered as variables. Studies of slipstreams have been conducted mainly in tunnels and laboratories. By performing a numerical analysis of the results of this study, the effects of the diffusion of air currents carrying air pollutants and viruses can be determined.

## 2. Materials and Methods

### 2.1. Experimental Equipment

A device measuring 3 m (W) × 35 m (L) × 2.5 m (H) was constructed to investigate the impact of the movement of an object on the slipstream in a large chamber. A rail was installed in the center of an indoor space to minimize the impact of the airflow. The mobile slipstream measuring device consisted of a rail, moving part, and sensor support (Patent-10-1517092). The rail consisted of a cylindrical aluminum pipe that helped the

moving part, which was a specially constructed motorcar, to move seamlessly at adjustable velocities (Figure 1).

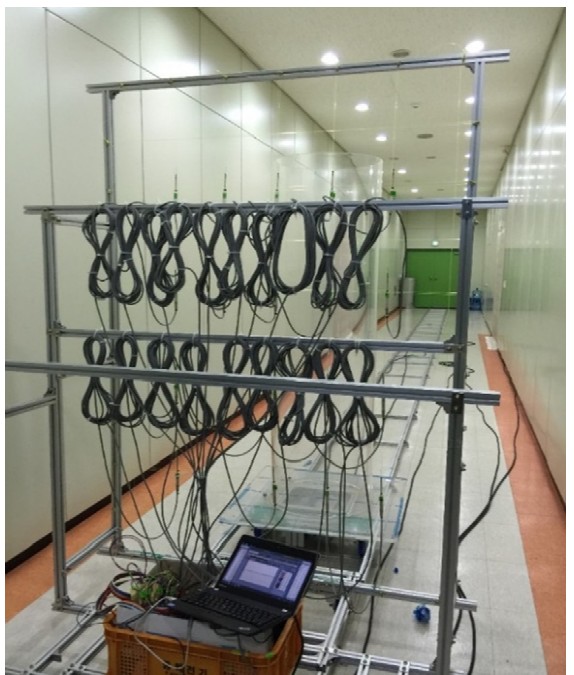 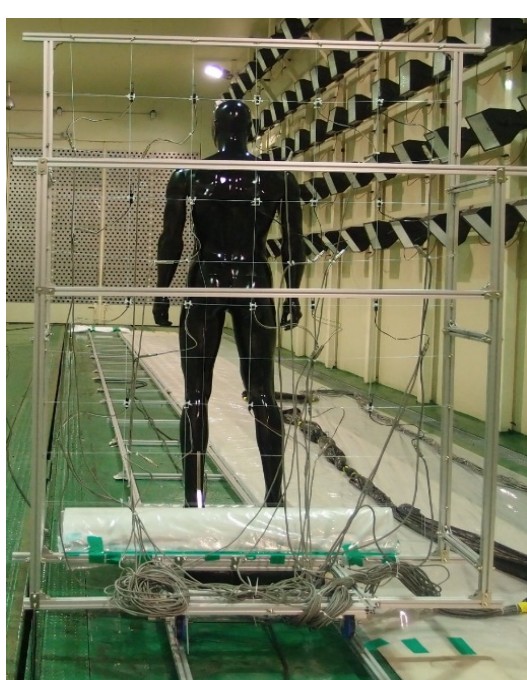

**Figure 1.** Testbed and sensor support.

A total of 20 wind velocity sensors (Model 0962-00, Kanomax, Andover, NJ, USA) were installed on the sensor support, with horizontal and vertical distances between sensors of 200 mm, as shown in Figure 2. The measuring range of the wind velocity sensors was 0.1–50 m/s, and the measurement error was ±0.1 m/s over the range 0–4.99 m/s. The relative wind speed was measured. The measured speed was subtracted from the human's moving speed to calculate the actual wind speed at each point.

$$U_r = U_h - U_m$$

Data were recorded every 0.1 s using a data logger (Model 1560, Kanomax, Suita, Japan). Table 1 shows the positioning of the sensors installed on the sensor support.

**Table 1.** Locations of wind velocity measurement sensors.

|  | Column B (X = −0.4 m) | Column C (X = −0.2 m) | Column D (X = 0.0 m) | Column E (X = 0.2 m) | Column F (X = 0.4 m) |
|---|---|---|---|---|---|
| 2nd Height (Y = 1.8 m) | B2 | C2 | D2 | E2 | F2 |
| 4th Height (Y = 1.4 m) | B4 | C4 | D4 | E4 | F4 |
| 6th Height (Y = 1.0 m) | B6 | C6 | D6 | E6 | F6 |
| 8th Height (Y = 0.6 m) | B8 | C8 | D8 | E8 | F8 |

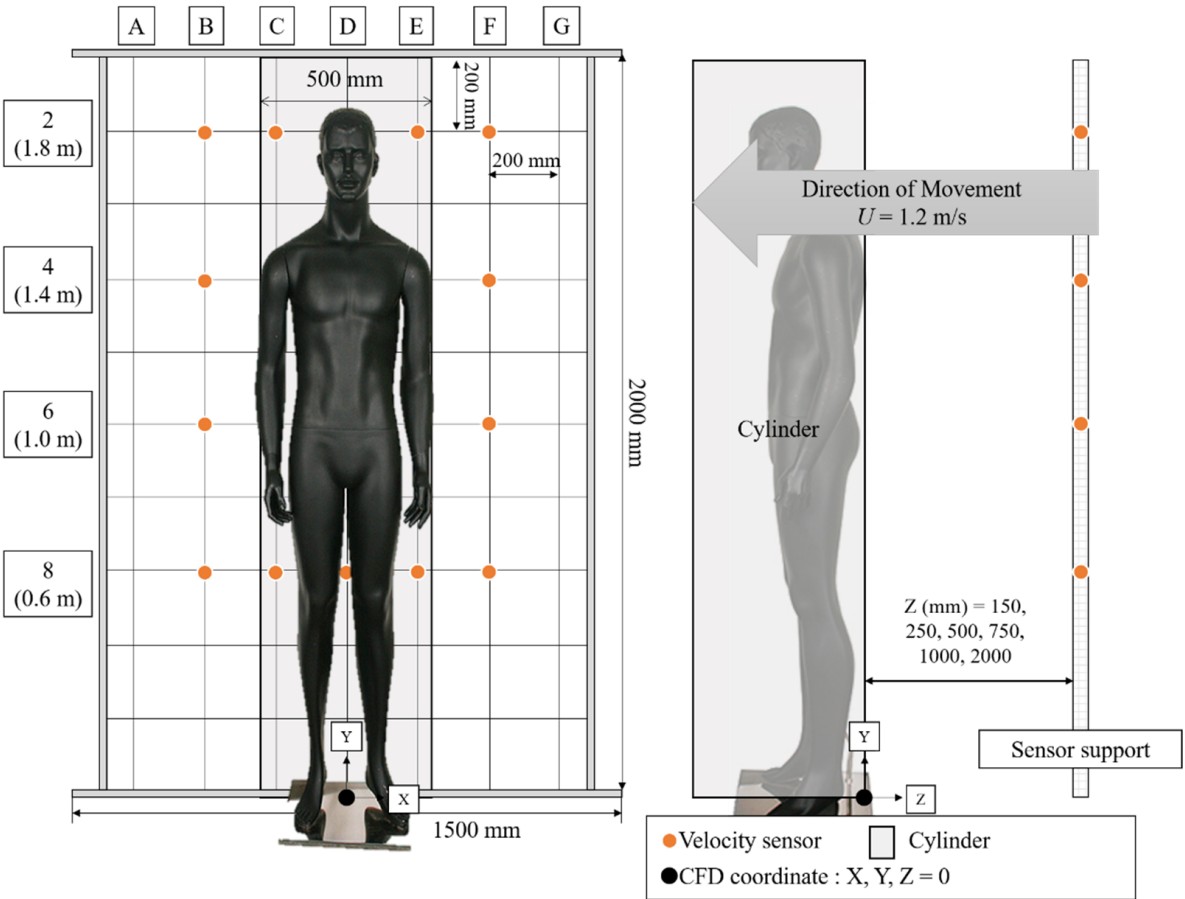

**Figure 2.** Schematic of the mobile wind velocity measurement system.

### 2.2. Experimental Methods

A normally shaped cylinder of diameter 500 mm, which is the average shoulder width of an adult male, was mounted on a motorcar and set to move at a velocity of 1.2 m/s. A running test was conducted five times, with a few pieces of corrugated cardboard placed in the empty spaces between the object and sensor support to flatten the bottom. Measurements were conducted at the rear of the cylinder after varying the sensor support position to 0.15, 0.25, 0.5, 0.75, 1, or 2 m. A numerical analysis was performed to evaluate the airflow of the cylinder slipstream. Fluent 17.7 software (Ansys Co., Canonsburg, PA, USA) was used to conduct the numerical analysis, and the test bed was modeled. The test bed dimensions were 3 m (W) × 2.5 m (H) × 10 m (L), and a cylinder of diameter 500 mm (2 m high) was placed at the center. Tetrahedral grids were placed in a tight formation around the cylinder and along the wall. A total of 354,330 nodes were generated in the flow field. The right, left, top, and bottom of the cylinder were specified as walls, and the no-slip condition was set. The front of the cylinder was specified as the inlet, and the wind velocity was set to 1.2 m/s (the walking speed of an adult male). The Reynolds number was $4.0 \times 10^4$. The rear surface was specified as the outlet and set to atmospheric conditions. A model built with the commercial code K-epsilon was used to solve the numerical analysis. The k-epsilon equation used in this study is as follows.

$$\mu_t = \frac{C_\mu \rho K^2}{\epsilon}$$

$$\rho \left( U_i \frac{\partial k}{\partial x_i} \right) = \frac{\partial}{\partial x_i} \left( \mu + \frac{\mu_t}{\sigma_k} \frac{\partial k}{\partial x_i} \right) + \mu_t \left( \frac{\partial U_i}{\partial x_j} + \frac{\partial U_j}{\partial x_i} \right) \frac{\partial U_i}{\partial x_j} - \rho \epsilon$$

$$\rho\left(U_i\frac{\partial\varepsilon}{\partial x_i}\right) = \frac{\partial}{\partial x_i}\left(\mu + \frac{\mu_t}{\sigma_\varepsilon}\frac{\partial\varepsilon}{\partial x_i}\right) + C_{\varepsilon 1}\frac{\varepsilon}{k}\left[\mu_t\left(\frac{\partial U_i}{\partial x_j} + \frac{\partial U_j}{\partial x_i}\right)\frac{\partial U_i}{\partial x_j}\right] - C_{\varepsilon 2\rho}\frac{\varepsilon^2}{k}$$

After a comparison of the results obtained using the cylinder, mannequin measurements were conducted using the same process.

## 3. Results and Discussion

### 3.1. Measurement Results

Each variable was measured five times using a mobile slipstream measuring device, and the average value was calculated. Figure 3 shows the measurement results at the 4th height position (Y = 1.4 m). In this case, Z was the distance between the cylinder and sensor support. The motor reached a speed of 1.2 m/s after 4 s and then continued at this velocity for approximately 8 s. The wind velocity approached the moving velocity of 1.2 m/s as Z increased. Although the change in wind velocity for rows B and F was not significant, the wind velocity near the sensor support was approximately 1.5 m/s, which was higher than the moving velocity of 1.2 m/s. The wind speed difference among rows B and F was not statistically significant according to non-parametric test results ($p > 0.05$). On the other hand, although the wind velocity varied greatly for rows C, D, and E according to the distance, the wind velocity near the sensor support was approximately 0.4 m/s, which was lower than the moving velocity of 1.2 m/s (Table 2).

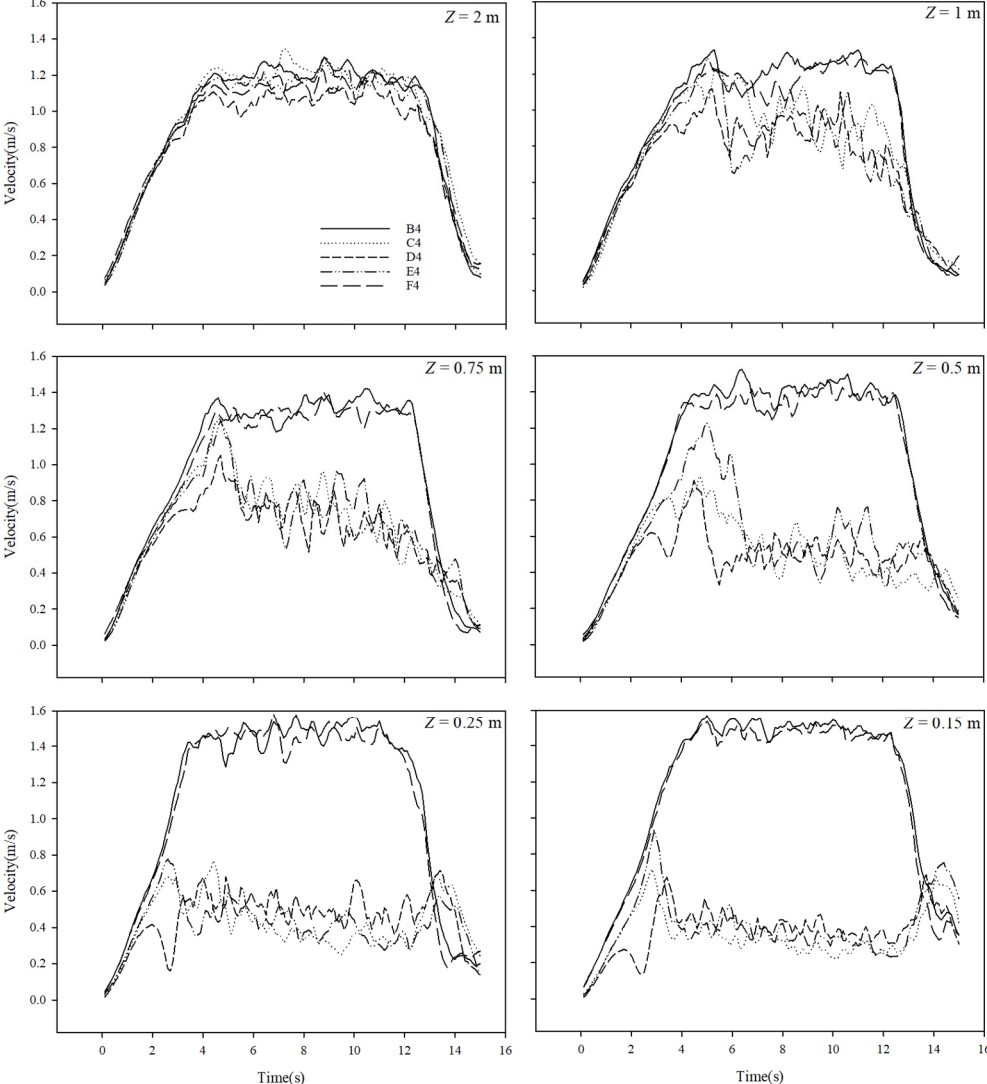

**Figure 3.** Velocity measurements at different locations (average of five measurements taken at the 4th height position).

**Table 2.** Wind velocity measurement results (average of five measurements taken at the 4th height position).

|  | Column B (X = −0.4 m) | Column C (X = −0.2 m) | Column D (X = 0.0 m) | Column E (X = 0.2 m) | Column F (X = 0.4 m) |
|---|---|---|---|---|---|
| Z = 2 m | 0.944 ± 0.027 | 0.870 ± 0.049 | 0.788 ± 0.037 | 0.886 ± 0.030 | 0.877 ± 0.047 |
| Z = 1 m | 1.003 ± 0.032 | 0.527 ± 0.089 | 0.553 ± 0.092 | 0.690 ± 0.119 | 0.901 ± 0.031 |
| Z = 0.75 m | 1.034 ± 0.042 | 0.396 ± 0.074 | 0.442 ± 0.095 | 0.466 ± 0.158 | 0.963 ± 0.029 |
| Z = 0.5 m | 1.108 ± 0.034 | 0.2450 ± 0.044 | 0.244 ± 0.024 | 0.299 ± 0.027 | 0.943 ± 0.085 |
| Z = 0.25 m | 1.158 ± 0.043 | 0.384 ± 0.040 | 0.385 ± 0.041 | 0.313 ± 0.033 | 1.108 ± 0.064 |
| Z = 0.15 m | 1.188 ± 0.040 | 0.311 ± 0.043 | 0.354 ± 0.038 | 0.237 ± 0.052 | 1.159 ± 0.024 |

Figure 4 shows the measurement results at various sensor heights taken at a distance of 0.5 m between the cylinder and sensor support. Similar patterns were observed to the right and left of row D. Wind velocity decreased as the sensor height decreased, which appeared to be because there was no significant difference in distance between the second height position and the top of the cylinder, although the sensor at the second height position was positioned behind the cylinder. A similar trend was observed in previous studies as the wind speed varied according to the measurement height [23,24].

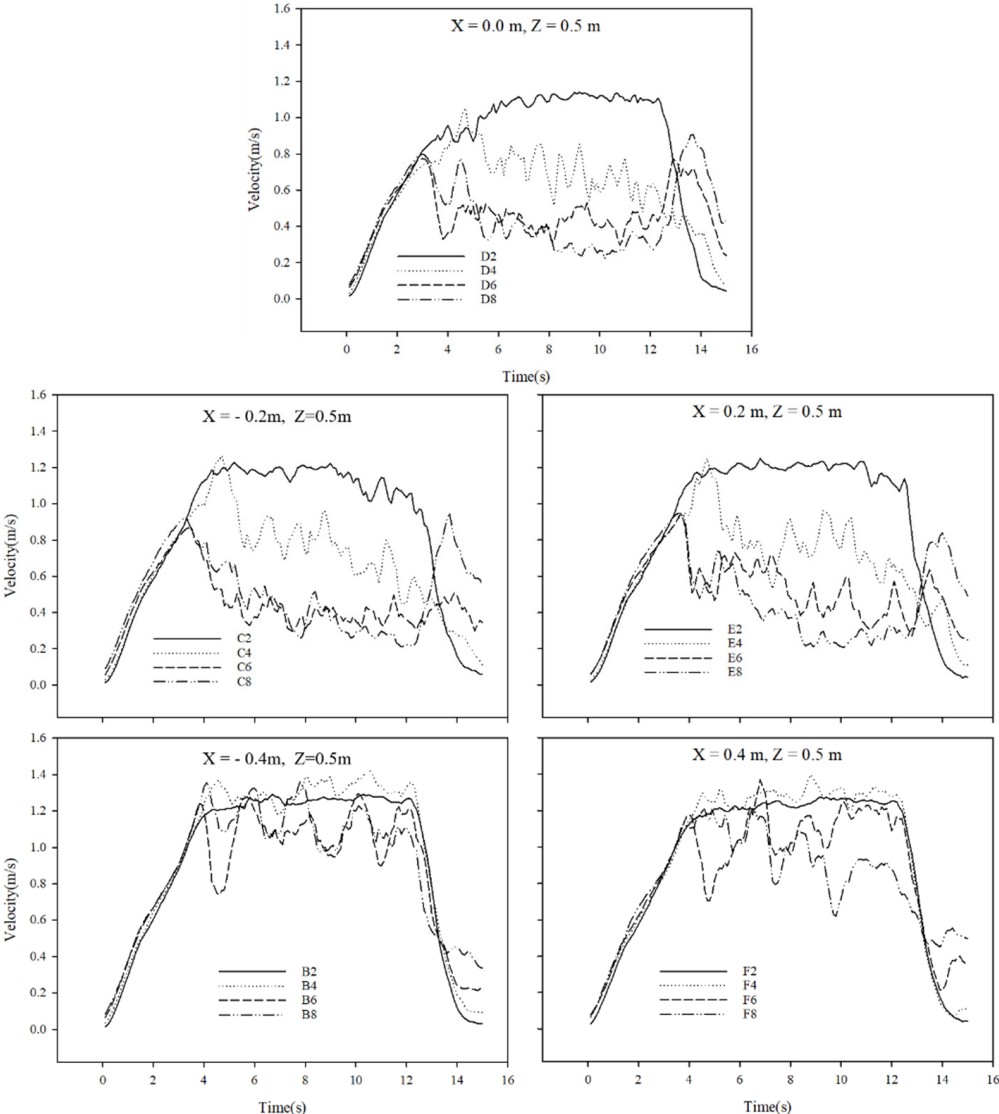

**Figure 4.** Velocity measurements at different locations (average of five measurements, Z = 0.5 m).

### 3.2. Results of Cylinder Measurements and the Numerical Analysis

The average Reynolds number in the 5–6 s constant velocity section was calculated using the measurement results presented above. Figure 4 shows the variation in the dimensionless velocity ratio calculated by dividing the measured velocity $u$ by the moving velocity $U$. Here, $X/d$ is a dimensionless value calculated by dividing the width of the cylinder by the diameter, and $Z/d$ is a dimensionless value calculated by dividing the distance between the cylinder and sensor support by the diameter of the cylinder.

Figure 5 shows that $u/U$ approached 1 as the airflow increased with the increase in the distance from the rear of the cylinder. This was also confirmed by the measurements made in this study. The value of $u/U$ at $X/d = 0$ increased toward 1 as $Z/d$ increased, and the right and left sections of the graph at $X/d = 0$ were symmetrical at all heights. The values of $u/U$ at $X/d = -0.4$, 0, or 0.4 were directly affected by the cylinder because sensor supports were positioned immediately behind the cylinder, and $u/U$ increased toward 1 as $Z/d$ increased.

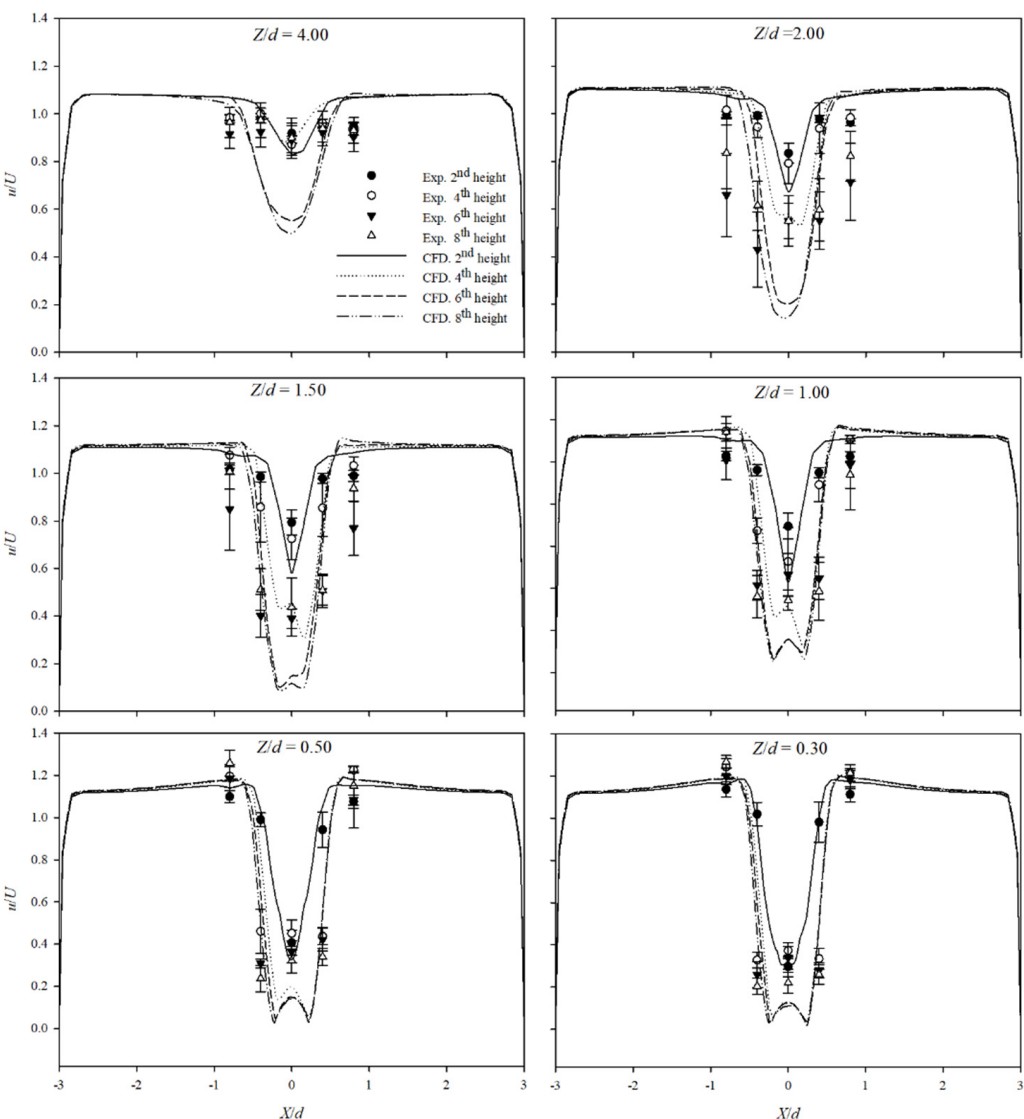

**Figure 5.** Profiles of normalized velocities at different $Z/d$ locations.

The results of the numerical analysis and the $u/U$ measurements are similar, with an approximate difference of only 0.023. This difference increased with $Z/d$, with the maximum difference (0.442) observed at the sixth and eighth height positions when $Z/d = 4$. However, the maximum difference at the second and fourth height positions was 0.089, which was not greater than that observed at the sixth and eighth height positions when $Z/d = 4$. This appeared to have been affected by the trailing vortex generated at the edge of the top of the cylinder [25].

Figure 6 shows the velocity profile according to $Z/d$ when $X/d = 0$. The numerical analysis of $U_x$ showed that the wind direction changed. The value of $u_x/U_x$ was –0.130 in the opposite direction of the wind at all heights when $Z = 0.15$ m, whereas the value of $u_x/U_x$ was –0.009 at the sixth and eighth height positions in the opposite direction of the wind for heights up to $Z = 0.75$ m. As shown in Figure 7, the value of $u_x/U_x$ was negative because a vortex was generated when the cylinder moved, and the vortex increased as the height decreased. It was presumed that the point where the vortex was generated was the point where the direction of the measured value changed.

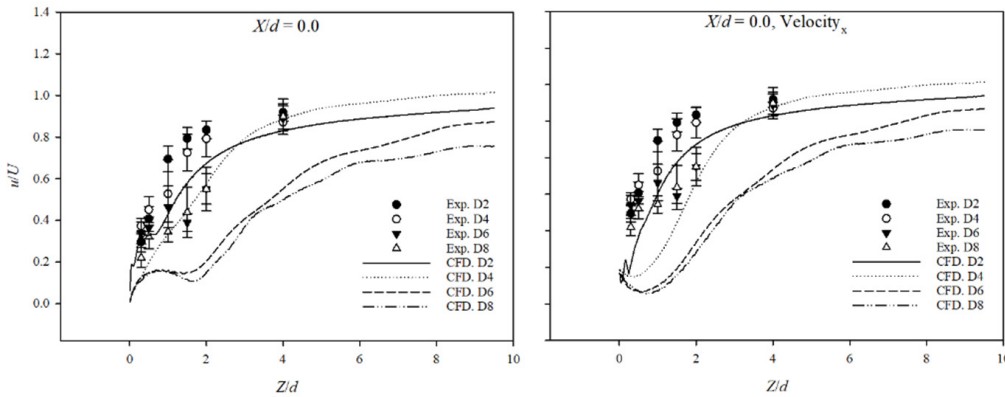

**Figure 6.** The normalized velocity profiles obtained by measurements and numerical analysis results.

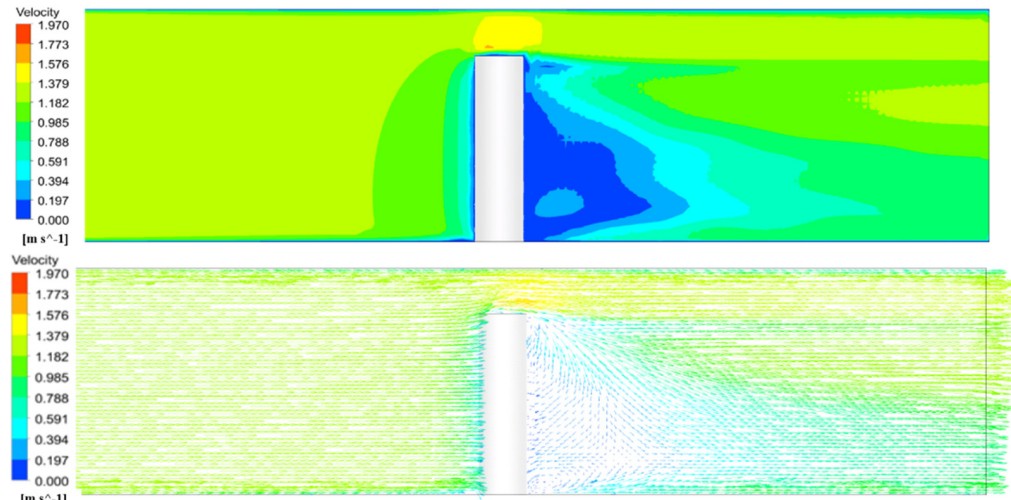

**Figure 7.** Velocity vector profile of the cylinder in the y–z plane (x = 0 m).

Figure 8 shows the fluid flow at four heights. It also shows that the velocity increased rapidly to the left and right of the cylinder, and that a vortex was created behind it. The strength of the vortex increased toward the bottom.

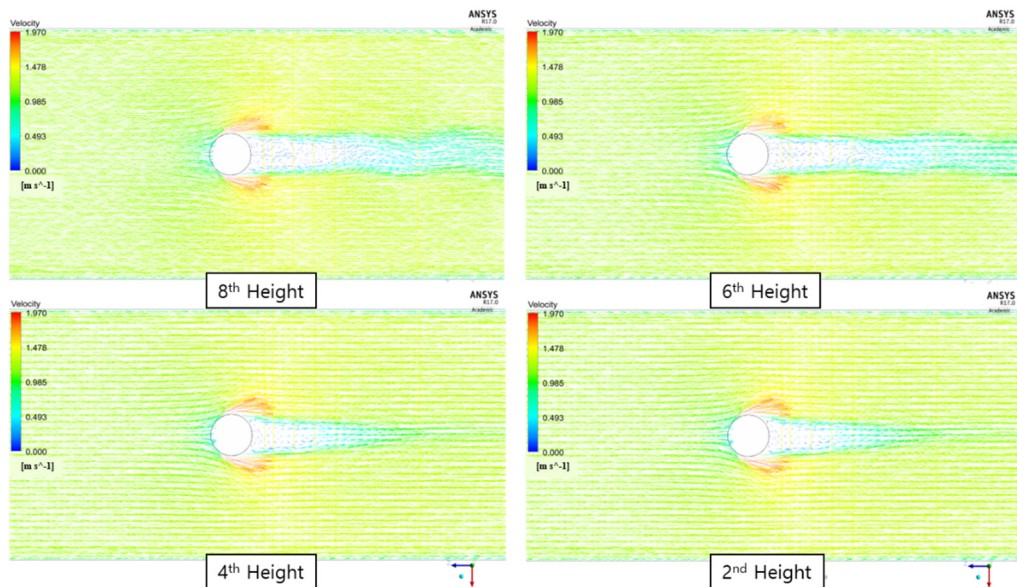

**Figure 8.** Time-averaged streamlines in different x–z planes.

Figure 9A shows the measurement results for the mannequin at the fourth height position (Y = 1.4 m). Figure 9B shows the measurement results according to various sensor heights taken at a distance of 0.5 m between the mannequin and sensor support. Patterns similar to those for Figure 9A,B of row D were observed. The average Reynolds number in the 5.6–8.0 s constant velocity section was calculated using the measurement results described above. Figure 10 shows the dimensionless velocity ratio calculated by dividing the measured velocity *U* by the moving velocity.

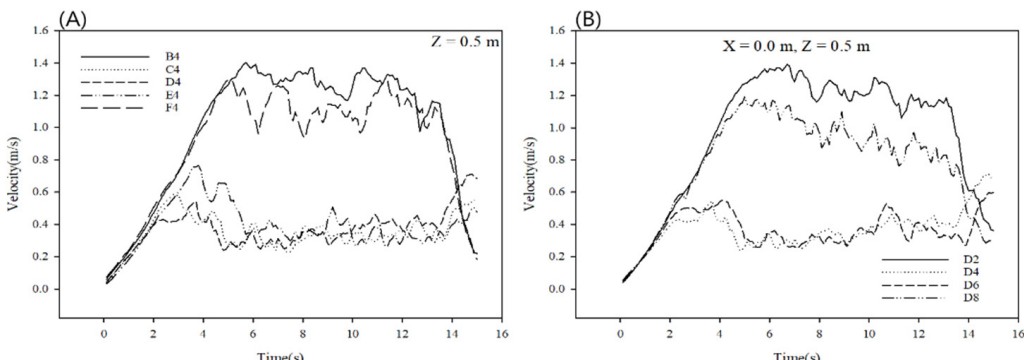

**Figure 9.** Results obtained at the 4th height position; (**A**) results for the mannequin at the fourth height position; (**B**) results according to various sensor heights taken at a distance of 0.5 m.

Figure 10 shows that as *u/U* approached 1, the air flow was more developed as the distance from the rear of the mannequin increased. This was also confirmed by the measurements in the study. The *u/U* at X (m) = 0 increased toward 1 as Z increased, and the right and left values at X (m) = 0 were symmetrical at all heights. The positions X (m) = −0.4, 0, or 0.4 were directly affected by the cylinder because they were positioned immediately behind the cylinder, and *u/U* increased toward 1 as *Z/d* increased.

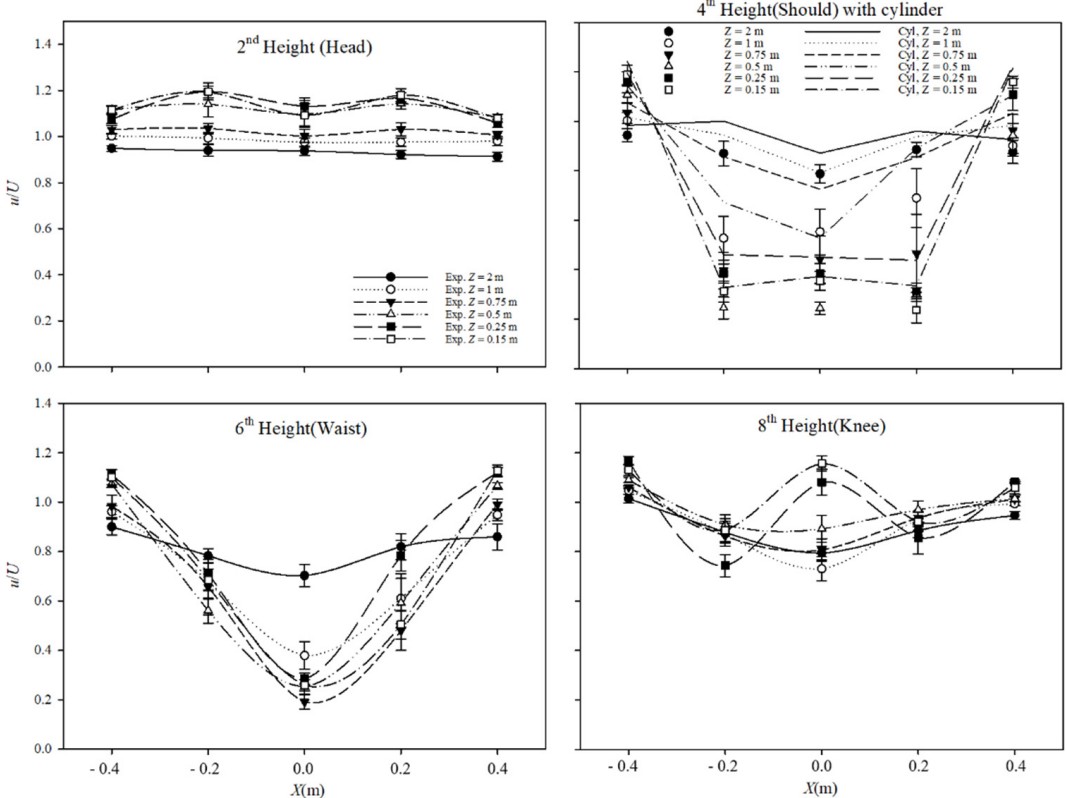

**Figure 10.** Results obtained for the cylinder and mannequin at the shoulder position (500 mm).

## 4. Conclusions and Discussion

In this study, a mobile measuring device was used to directly measure the slipstream generated when a cylinder of diameter 500 mm (the average shoulder width of an adult male) was moved at a velocity of 1.2 m/s (the average walking velocity of humans) to obtain the velocity profile of the rear side of a cylinder. The cylinder was then fixed, and a wind velocity of 1.2 m/s was created at the inlet to provide data for the numerical analysis; the results were compared with the measurements. The results of the measurements and numerical analysis of the area behind the cylinder are very similar, with a maximum $u/U$ difference of 0.066 at $Z/d = 0.3$. The difference was 0.042 and was larger at the 2nd height position when $X/d = 0.0$ and $Z/d = 4$. The increase in the difference with increasing $Z/d$ was attributed to the impact of the trailing vortex, and to the fact that the numerical analysis was performed with the cylinder in a fixed position. The direction of the measurement was estimated as per the position of the vortex obtained from the numerical analysis. The wake of the mannequin was measured in the same way as that of the cylinder. Similar measurement results were obtained at the shoulder height, which was the same diameter as the cylinder, and a velocity profile for the mannequin wake was obtained.

Previous experimental studies [20] on wakes were conducted in wind tunnels or as lab-scale experiments, whereas the present study used a mobile cylinder that can reveal the actual characteristic of wakes. In this manner, air flow characteristics were identified. This study presented a method based on a mobile measuring device to obtain the velocity profile of the slipstream of a moving object. If the moving velocity of the sensor support is accounted for, the mobile measuring device could be used to verify the results of simulations of the dispersion of airborne contaminants derived from moving objects.

Recently, the social and economic damage caused by viruses such as COVID-19 has increased. To prevent the spread of such airborne substances, a ventilation system can be used, along with restrictions on movement [26,27]. Droplet dispersion during coughing by people who are walking plays an important role in the transmission of COVID-19. A

simulation showed that droplets in the air in a narrow space, such as a hallway, were transmitted below waist height (of the emitter) [28]. Wake measurements confirmed that the speed and distribution stabilized as the distance from the mobile slipstream at chest and waist height increased. The speed rapidly increased to the left and right of the mobile slipstream, and a vortex was created on the back side. The vortex formed as it moved to the bottom surface. Based on the results of this study, movement restrictions appear necessary to slow the spread of the virus, in accordance with the correlation between the wake velocity distribution and spread of the virus.

Numerical analysis results for a system comprising a moving object could not be compared with the measurement results. Therefore, in future studies, methods need to be devised that consider moving velocity when measuring wind velocity, and numerical analysis of a system including a moving object could be performed for comparison.

**Author Contributions:** D.P. planned the study and contributed the main ideas; M.K. was principally responsible for the writing of the manuscript; Y.L. revised the manuscript and methodology analysis. All authors have read and agreed to the published version of the manuscript.

**Funding:** This work was supported by a grant from the Subway Fine Dust Reduction Technology Development Project of the Ministry of Land Infrastructure and Transport (21QPPW-B152306-03).

**Conflicts of Interest:** The authors declare no conflict of interest.

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
