# Peer review of "Analysis of the Airflow Generated by Human Activity Using a Mobile Slipstream Measuring Device"

_environments, doi:10.3390/environments8100097_

Round 1
Reviewer 1 Report
This study is devoted to investigating the airflow velocity of human activity by using a mobile device. The idea raised from this study is interesting, however, there are some serious flaws in the statistical analysis. Throughout the manuscript, generalizations are used without conducting the statistical comparisons. In addition, I think the data in this article is quite complete, but it lacks in-depth discussion. Therefore, this manuscript in its current form is not fit for publication. Several recommendations are given as follows:
- Line 34, needs a reference. At the same time, this sentence is almost the same as line 70.
- Line 131, the insignificant difference needs a correct statistical method to prove.
- Also, lines 132-133, and line 135. The difference between two different wind velocities still has to be verified by a correct statistical method.
- Line 144, please exactly describe what’s similar findings discovered from these two studies.
- Authors emphasize measuring the velocity profile of a moving object is important since this information is related to the contamination disperse in the environment. However, in my opinion, no in-depth discussion is described in this manuscript. Please explain the possible correlation between the results of these measurements and the spread of contamination (or virus)? If these contaminations are spread out at such a speed, how far is the range of possible impact? Based on this result, the author believes that restrictions on movement are effective or unnecessary?
Reviewer 2 Report
In this paper a study of the analysis of generated airflow according to human activity using a mobile slipstream measuring device is made. Before being published, I suggest some improvements in the presentation and in the content.
The introduction should be improved and more references should be added.
More details about the experimental setup and equipment should be improved.
More details about the numerical model should be introduced, as Equations, grid generation,…
In figure 6 and 7 should be improved. More details about the recirculation area should be presented. More conclusions about the correlation of the airflow and the spread of various airborne substances such as COVID-19 should be added.
Round 2
Reviewer 1 Report
1.Line 149: Although the change in wind velocity for rows B and F was not significant. Authors need to use nonparametric statistics to confirm whether the data are significantly or insignificantly different.
2. Please make sure that every equation present in the text is clear
Reviewer 2 Report
In the actual version, in general, all suggestions given by the reviewer was implemented.
Author Response
Please see the attachment.

This manuscript is a resubmission of an earlier submission. The following is a list of the peer review reports and author responses from that submission.